# The Functional Association of *ACQOS/VICTR* with Salt Stress Resistance in *Arabidopsis thaliana* Was Confirmed by CRISPR-Mediated Mutagenesis

**DOI:** 10.3390/ijms222111389

**Published:** 2021-10-21

**Authors:** Sang-Tae Kim, Minkyung Choi, Su-Ji Bae, Jin-Soo Kim

**Affiliations:** 1Department of Medical & Biological Sciences, The Catholic University of Korea, Bucheon 14662, Korea; 2Center for Genome Engineering, Institute for Basic Science, Daejeon 34126, Korea; mkchoi@ibs.re.kr (M.C.); sjbae@ibs.re.kr (S.-J.B.); jskim01@snu.ac.kr (J.-S.K.)

**Keywords:** CRISPR, Cas9, guide RNA, ACQOS, Arabidopsis, salt stress, chlorophyll

## Abstract

Clustered regularly interspaced palindromic repeat (CRISPR)-mediated mutagenesis has become an important tool in plant research, enabling the characterization of genes via gene knock-out. CRISPR genome editing tools can be applied to generate multi-gene knockout lines. Typically, multiple single-stranded, single guide RNAs (gRNAs) must be expressed in an organism to target multiple genes simultaneously; however, a single gRNA can target multiple genes if the target genes share similar sequences. A gene cluster comprising ACQUIRED OSMOTOLERANCE (*ACQOS*; *AT5G46520*) and neighboring nucleotide-binding leucine-rich repeats (*NLR*s; *AT5G46510*) is associated with osmotic tolerance. To investigate the role of *ACQOS* and the tandemly arranged NLR in osmotic tolerance, we introduced small insertion/deletion mutations into two target genes using a single gRNA and obtained transformant plant lines with three different combinations of mutant alleles. We then tested our mutant lines for osmotic tolerance after a salt-stress acclimation period by determining the chlorophyll contents of the mutant seedlings. Our results strongly suggest that *ACQOS* is directly associated with salt resistance, while the neighboring NLR is not. Here, we confirmed previous findings suggesting the involvement of *ACQOS* in salt tolerance and demonstrated the usefulness of CRISPR-mediated mutagenesis in validating the functions of genes in a single genetic background.

## 1. Introduction

The development of RNA-guided genome editing tools, such as clustered regularly interspaced palindromic repeat (CRISPR)-associated (Cas) protein systems, has led to technical advances in targeted mutagenesis in plants [1,2,3,4,5,6,7,8,9,10]. CRISPR-Cas systems enable highly accurate genome editing using sequence-specific single-stranded single guide RNA (gRNA), the first 20 nucleotides of which are typically designed to match the target DNA sequence. Cas proteins, which comprise DNA binding and nuclease domains, introduce double-strand breaks 3 bp upstream of protospacer adjacent motifs (PAMs); for example, the PAM recognized by Cas9 isolated from *Streptococcus pyogenes* (*Sp*Cas9) is 5′-NGG-3′. Using this mechanism, short insertion or deletion (in/del) mutations can be introduced at the target DNA site via endogenous DNA repair, homology-directed recombination, or non-homologous end joining [11,12]. Although longer in/del mutations are possible [13], CRISPR/Cas9 typically introduces short in/del mutations at the target site; this can be applied to introduce frame-shift mutations in target genes, resulting in the production of a nonfunctional gene product. Furthermore, CRISPR-based genome editing tools that function without introducing a double-strand break have been developed, such as base editor [14,15,16,17,18,19] and prime editor [20,21]. CRISPR-mediated genome editing techniques are fast becoming the standard tools used for genetic and physiological analyses, functional genomics, and crop improvements in plants.

Polyploidy and hybridization commonly occur during plant evolution; these mechanisms contribute to species diversity and the presence of redundant genes in plants [22,23]. Polyploidization via hybrid speciation, as well as whole genome duplication, have led to the acquisition of gene paralogs in various plant species. Thus, mutagenesis techniques that can target multiple gene paralogs or genes belonging to families with high sequence homology simultaneously are needed to investigate genic functions in plants. The conventional method for generating plant lines with multiple gene knockouts is to artificially cross two single-mutant lines. CRISPR/Cas9-mediated mutagenesis enables several mutations to be introduced simultaneously, typically by utilizing multiple gRNAs to actively target multiple sites.

Ribonucleoprotein-mediated CRISPR/Cas9 delivery for genome editing is possible in plant systems [24]; however, due to the challenges associated with producing whole regenerated plants, *Agrobacterium*-mediated transformation is commonly used to introduce binary vectors into plants for gRNA and Cas9 expression. Several methods for simultaneously expressing multiple gRNAs targeting different sites are available. Multiple gRNAs can be expressed using a different promoter and terminator for each gRNA, or by expressing RNA-cleaving enzymes such as 5′-end hammerhead and 3′-end hepatitis delta virus dual ribozyme, CRISPR-associated RNA endoribonuclease Csy4 from *Pseudomonas aeruginosa*, or tRNA processing enzymes [20,25,26,27,28,29,30]. Furthermore, PAM positioning must be considered when designing multiplexed experiments involving sgRNAs targeting multiple specific sites. Notably, a single gRNA can be used to target multiple DNA targets whose sequences are identical. Thus, it might be an acceptable strategy to design one specific gRNA targeting multiple genes simultaneously, although there is a practical difficulty in that this strategy will be useful only when the PCR product for each target site can be uniquely amplified for genotypically assessing the genome editing efficiency.

Previous research examining natural genetic variation in the model plant species *Arabidopsis thaliana* have provided insight into gene functions and trait evolution associated with adaptation and selection [31,32,33]. The osmotic stress tolerance of several natural *A. thaliana* accessions was compared after subjecting the plants to an acclimation period with mild salt stress [34]. Recently, a novel locus associated with salt tolerance, acquired osmotolerance (*ACQOS*), was identified based on genome-wide association mapping and then characterized using T-DNA knockout lines [35].

The *ACQOS* locus is a cluster comprising four nucleotide-binding leucine-rich repeats (NLRs): *AT5G46490*, *AT5G46500*, *AT5G46510*, *AT5G46520* (*ACQOS*). *ACQOS* is identical to VARIATION IN COMPOUND TRIGGERED ROOT GROWTH RESPONSE (*VICTR*; *AT5G46520*), which encodes a toll-interleukin1 receptor-nucleotide-binding leucine-rich repeat class protein [36]. *ACQOS* and an upstream NLR (*AT5G46510*) in the wild-type *A. thaliana* ecotype, Col-0 are paralogs that likely resulted from a recent duplication event. Plants harboring wild-type *ACQOS* alleles are sensitive to salt stress after acclimation. However, plants containing full deletions or pseudogenization-induced polymorphisms in *ACQOS* and *AT5G46510* show considerable tolerance to salt stress, suggesting that *ACQOS* suppresses osmotic tolerance [35].

In this study, we performed CRISPR/Cas9-mediated mutagenesis in *A. thaliana* wild type lab line, Col-0 to target multiple genes using a single specific sgRNA and then characterized the mutant phenotypes. We established single and double knockout lines for *ACQOS* alleles and measured the chlorophyll content of the transgenic seedlings exposed to salt stress (Figure 1A).

## 2. Results

### 2.1. Generating Mutant Lines by Targeting Two Genes Using One Specific sgRNA for CRISPR-Mediated Mutagenesis in Arabidopsis

To generate CRISPR-mediated knockout *Arabidopsis* transformants, we used *Agrobacterium* to transform the binary CRISPR/Cas9 vector pBAtC [11] into *Arabidopsis* for SpCas9 and sgRNA expression. We designed an sgRNA (5′-AAGAGTAGAGAAACTTACAA-3′) that could target two genes simultaneously, *AT5G46510* and *ACQOS*. In/del mutation efficiencies were measured by amplicon deep sequencing; for details, see Appendix A. The in/del mutation efficiencies in 12 individual plants of the first generation after transformation (T_1_) ranged between 0.6% and 99.4% for *AT5G46510* and 0.8% and 99.5% for *AC**QOS* (Figure 1B). T_1_ plant S-10 had the highest in/del mutation rates (99.4% for *AT5G46510* and 99.5% for *ACQOS*); thus, seeds from S-10 were harvested and germinated.

The in/del mutation efficiencies were then assessed in 38 s-generation (T_2_) plants by amplicon deep sequencing. The mutation efficiency ranges in the T_2_ plants, which were calculated using the in/del ratios, were 0.05–100% and 0.0–99.98% for *AT5G46510* and *ACQOS*, respectively (Figure 1C). Traces of the T-DNA vector transformed into the T_1_ plant by *Agrobacterium* were detected via PCR using primers specific for Cas9 and herbicide resistance genes; no vector traces were found in 10 of 38 T_2_ individuals (Figure 1C, Appendix A). Seeds were harvested from T_2_ plants 6, 7, 12, and 30 to generate the three possible allelic variations in third generation (T_3_) plants: homozygous mutations in *AT5G46510* and wild-type *ACQOS*, wild-type *AT5G46510* and homozygous mutations in *ACQOS*, and homozygous mutations in both *AT5G46510* and *ACQOS*. T_2_ plant 6 had in/del ratios of 0.4% for *AT5G46510* and 45.5% for *ACQOS* and did not harbor traces of the T-DNA plasmid (Figure 1C and Appendix A). T_2_ plant 30 exhibited in/del ratios of 63.62% and 2.02% for *AT5G46510* and *ACQOS*, respectively, with no T-DNA traces. T_2_ plants 7 and 12 had almost 100% in/del frequencies in both genes: 99.69% for *AT5G46510* and 99.88% for *ACQOS* in T_2_ plant 7, and 99.91% for *AT5G46510* and 99.91% for *ACQOS* in T_2_ plant 12 (Figure 1C). T_2_ plants 7 and 12 contained traces of T-DNA (Figure 1C).

Next, the mutation ratios and allele compositions of T_3_ plants from parental T_2_ plants 6, 7, 12, and 30 were measured by amplicon deep sequencing (Figure 1A and Figure 2A). All 12 T_3_ individual progeny of T_2_ plant 6 had wild type *ACQOS* alleles; however, four individuals had homozygous, and four had heterozygous mutations in *AT5G46510* (Figure 1D). Similarly, all 12 T_3_ individuals derived from T_2_ plant 30 had wild type *AT5G46510* alleles, while 6 individuals had homozygous mutations and 5 heterozygous mutations in *ACQOS* (Figure 1D). The alleles in the T_3_ individuals were inherited from the T_2_ parent plants (6 and 30); no T-DNA was present in any of the T_3_ plants (Figure 1A). Although two T_3_ individuals varied in their mutation rate (in/del frequencies in T_3_ 7-08 and 7-09) most T_3_ plants derived from T_2_ plants 7 and 12 carried homozygous mutations in *AT5G46510* and *ACQOS* (Figure 1D). Only one individual did not carry traces of T-DNA and was hence selected for the downstream experiments (Appendix A). Six T_3_ plants were used to generate fourth generation (T_4_) plants for characterization: two T_3_ plants with homozygous mutations in *AT5G46510* and wild-type *ACQOS* (M/W; from T_2_ line 30), two T_3_ plants with wild-type *AT5G46510* and homozygous mutations in *ACQOS* (W/M; from T_2_ line 6), and two T_3_ plants with homozygous mutations in both *AT5G46510* and *ACQOS* (M/M; from T_2_ lines 7 and 12).

### 2.2. Salt Resistance Responses with Four Different Sets of Allelic Mutants

Our successful T_3_ lines are expected to produce defective, non-functional proteins, with mutant alleles as illustrated in Figure 2A. Thus, four sets of T_4_ seeds harboring allelic mutants representing wild-type *AT5G46510* and mutant *ACQOS* (W/M parental lines; s6-1 s6-3), mutant *AT5G46510* and mutant *ACQOS* (M/M parental lines; s7-1, s12-7), mutant *AT5G46510* and wild-type *ACQOS* (M/W parental lines; s30-2, s30-3), and both wild-type *AT5G46510* and *ACQOS* (W/W Col-0) were germinated, and the responses of the seedlings to salt stress were examined (Figure 2A). Twelve individuals from each genetic line with four different allelic combinations in two genes were used per experiment, and three technical replicates were measured in two different experiment times. Seeds were sown on moisture filter papers soaked in Murashige and Skoog (MS) medium.

After 14 days, seeds were mostly germinated and grown into the seedlings for following physiological test. The 14-day-old seedlings were transferred to MS medium containing sodium chloride for short-term acclimation as described in the previous experiments [34]. After additional 7 days, short-term acclimated seedlings were again transferred into the plant medium containing sorbitol and incubated for 7 days more. Chlorophyll was then extracted by pooling approximately 12 seedlings of each line and measured as an indicator of plant salt resistance (Figure 2B).

Our chlorophyll measurements suggest that *ACQOS* silencing significantly affected salt stress tolerance; four of the plant lines with the W/M (wild-type *AT5G46510* / mutant *ACQOS*) and M/M (mutant *AT5G46510*/mutant *ACQOS*) genotypes exhibited chlorophyll contents that were comparable with those of wild-type Col-0 that is intolerant to salt stress. The chlorophyll contents were significantly lower in plants with the W/W and M/W genotypes compared with the other plant genotypes examined (*p* < 0.0001, one-way ANOVA with post hoc Tukey HSD test). When we compared our mutant lines to Bur-0, an *Arabidopsis* natural accession known to be salt stress resistant in the previous experiments with same stress conditions [34,35], W/M and M/M genotypic lines only showed no significant differences (Figure 2C). To exclude possible artifacts such as possible individual growth rate or maternal effects, the chlorophyll contents of treated seedlings were normalized to those of untreated seedlings. The relative chlorophyll contents were significantly different (*p* < 0.0001, one-way ANOVA with post hoc Tukey HSD test) in lines with the W/W (Col-0) and M/W (s30-2, s30-3) genotypes compared with the M/M (s7-1, s12-7) and W/M (s6-1, s6-3) genotypes (Figure 2D).

## 3. Discussion

In this study, we successfully performed CRISPR-mediated mutagenesis using a single specific gRNA to target two genes simultaneously. We induced in/del mutations in aiming two genes with one specific gRNA, which could generate three different allelic combinations, (1) wild type *AT5G46510* and mutant *ACQOS*, (2) mutant *AT5G46510* and wild type *ACQOS*, and (3) mutant *AT5G46510* and mutant *ACQOS*. These different mutant lines whose mutant alleles were inherited after additional generations and effectively utilized for physiological test on osmotic tolerances to salt-stresses as a proof of concept.

Our strategy was very efficient to generate mutant lines with allelic combinations on this tandemly arranged redundant genes. In particular, by initial tracking of editing efficiencies in T_1_ plant individuals using amplicon deep sequencing via NGS, we could find that one out of 12 T_1_ individuals exhibited high in/del efficiencies in both genes (*AT5G46510* and *ACQOS*). The range of somatic cell mutation ratios in the T_1_ plants in our study was similar to those in previous reports where the same vector was used. Approximately 10% of T_1_ individuals exhibited mutation ratios of >90% [11]. For the T_2_ generation, we determined the number of progenies derived from the highly mutated T_1_ line (99.4% for *AT5G46510* and 99.5% for *ACQOS*) with inherited mutations by detecting traces of the T-DNA vector and measuring the mutation ratio. Notably, most T_2_ individuals with high mutation ratios (>90% for *AT5G46510* and *ACQOS*) had traces of T-DNA; however, we found several T_2_ individuals with heterozygous mutant alleles for *AT5G46510* and *ACQOS* that carried no traces of T-DNA. The T_2_ plants with homozygous mutant alleles of both genes maintained the T-DNA vector, which indicates that most T_2_ plants with high rates of homozygous mutant alleles were induced by *de novo* genome editing via active Cas9 and gRNA in somatic cells, although we cannot rule out the possible inherited homozygous mutant alleles. We obtained inherited homozygous mutant lines for *AT5G46510* only, allelic combination of (2), and *ACQOS* only, allelic combination of (1), in the T_3_ generation using T_2_ plants with inherited heterozygous mutant alleles (Figure 1D); 12 T_3_ individuals exhibited Mendelian segregation for each gene. We generated one inherited homozygous mutant line in which both genes were mutated; however, most T_3_ plants still contained traces of T-DNA, indicating a possible induction of *de novo* mutations. The T_3_ lines from two parental T_2_ lines mostly carried homozygous mutant alleles in *AT5G46510* and *ACQOS*, suggesting that the homozygous mutant alleles of the T_3_ lines were inherited [9].

Strategies to obtain combinational knockout alleles of two genes via CRISPR-mediated mutagenesis include using a separate gRNA to target each gene, using a single gRNA to target both genes simultaneously, and crossing two individual mutant lines. Some advantages to our strategy, i.e., using a single gRNA to target two genes, include faster preparation of mutant lines compared with obtaining different allelic mutant lines by crossing between single mutant lines and reduced the complexity of gRNA design compared with using multiple gRNAs. The efficiencies of various promoters used for Cas9 expression during CRISPR-mediated genome editing have been assessed [27,28,37,38,39,40]. Using more efficient promoters, such as RPS5a, may enhance mutation rates in the T_1_ generation, thus increasing the chance of obtaining more inherited mutations in the T_2_ generation.

Here, we generated T_4_ seedlings with three different combinations of homozygous mutant and wild-type alleles in two tandemly arranged NLRs. Ariga et al. [35] strongly suggested that *ACQOS* plays a suppressive role in osmotic tolerance using GWAs with 179 natural accessions as they characterized 5 haplogroups at the *ACQOS* locus, which are different in *AT5G46510* and *ACQOS* (both genes are intact; haplogroup 1, nonfunctional *ACQOS*; haplogroup 2, *ACQOS* missing; haplogroup 3; or both genes are missing and two different types in *AT5G46490*-like *NLRs* (haplogroup 4 and 5)). Except accessions belonging to haplogroup 1, all other accessions showed salt resistance, but salt sensitivity was recovered by complementary test in the *ACQOS* knock-out line, *acqos*, by introducing Col-0 *ACQOS*. Our results support previous findings that *ACQOS* is associated with salt resistance as it is related to suppression of osmotic tolerance [35]. Chlorophyll content was measured as an indicator of osmotic tolerance under salt stress, and our data suggested that the loss-of-function *ACQOS* mutants were resistant to osmotic stress. The insertion of 1 bp in/del mutations via CRISPR-mediated genome editing effectively generated loss-of-function mutants by introducing frame shift mutations. This was a proof-of-concept study, in which multiple gene targets were mutated using CRISPR/Cas9 and a single gRNA, and then the phenotypes of the mutant lines were characterized by examining their salt stress responses. In particular, we could generate the *AT5G46510*-only nonfunctional allele, which is uniquely tested in this study and strongly confirmed that *ACQOS* is only one gene associated to the suppression of osmotic tolerance in this locus. We created mutant lines using the same genetic background, which more clearly enabled evaluation of the functions of *AT5G46510* and *ACQOS*.

CRISPR-mediated mutagenesis for the introduction of in/del mutations and base editing is a fundamental tool that can be used to mimic naturally occurring genetic variations. Precision genome editing is a powerful tool that can be used to confirm the functions of genes identified in genome-wide association studies and for characterizing the phenotypic traits found in natural plant variants.

## 4. Materials and Methods

### 4.1. Guide RNA Target Design

To design gRNA to target two different genes (*AT5G46510* and *AT5G46520*; *ACQOS*) for CRISPR/Cas9 mutagenesis, we initially selected three candidate sites in the second exon (Figure 2A) based on aligned nucleotide sequences of two genes obtained from The Arabidopsis Information Resource (TAIR) using the multiple sequence alignment program, MUSCLE [41]. All suggested gRNAs are unique when we tested whether there was identical sequences in the reference genome sequence of Col-0 using Cas-Offinder [42] implemented in rgenome.net (accessed on 21 October 2021, http://rgenome.net). From in silico analysis using Cas-Offinder software with the gRNA (5′-AAGAGTAGAGAAACTTACAA-3′) that we kept in this study, there appeared two potential off-target sites whose sequences contained three mismatched nucleotide sequences to our selected gRNA. This indicated that the selected gRNA is unique and suitable to target two sites designated for CRISPR-mutagenesis.

### 4.2. Vector Construction and gRNA Cloning

The CRISPR vector for this study was used in a previously published pBAtC [11] that is composed of the U6 promoter from Arabidopsis to express gRNA including CRISPR RNA scaffold, the 35S promoter originally from cauliflower mosaic virus (CaMV) to express Cas9 and the Basta^TM^ (Phosphinothricin) resistance gene for the selection of the first generation of transformants. More information of pBAtC is available in the databases, GenBank (KU213970) and addgene (addgene.org; #78097). To clone the guide RNA in the vector, we digested vectors with the restriction enzyme, *AarI* (ThermoFisher Science, Waltham, MA, USA) for 3 h to produce linearized pBAtC with 4-bp overhangs. Then, we mixed both annealed products of two oligonucleotides synthesized representing the gRNA target site and T4 DNA ligase (New England Biolabs, Ipswich, MA, USA) for ligation. Forward and reverse oligos for the guide sequence were diluted with T4 DNA ligase 10X buffer (New England BioLabs^®^, Cat# B0202S) to 10 μM each and annealed using the following thermocycling program: pre-denaturation for 1 min at 95 °C; 95–25 °C for –1 °C/min; and hold at 10 °C. After ligation, plasmids were purified using the ExpinTM PCR SV mini kit (GeneAll^®^, Cat# 103-202) and transformed into *E*. *coli* competent cells, as previously described [11]. *E*. *coli* transformants were selected on spectinomycin; plasmid DNA was isolated using the ExprepTM plasmid SV mini kit (GeneAll^®^, Cat# 101-102) and validated by Sanger sequencing. All primers used in this study are listed in Appendix A.

### 4.3. Agrobacterium-Mediated Transformation

The *Arabidopsis thaliana* Col-0 was utilized for *Agrobacterium tumefaciens* GV3101-mediated transformation following the method of floral dip [43]. The growth condition for *Arabidopsis* plants was the long day condition (16 h light/8 h dark) at 22 °C in a growth room (Koencon, Hanam, South Korea) with light of 32 W Osram lamp (170 mmol/2 m/s).

### 4.4. Targeted Deep Sequencing and Mutation Pattern Analysis

We extracted genomic DNA for targeted deep sequencing analysis via NGS from two or more randomly selected leaves from each plant individual to avoid possible bias of mutation rates in chimeric T_1_ plants using a commercially available plant DNA extraction kit. The PCR product for the target sequence was amplified from genomic DNA using target-specific primers that were designed using reference genome sequences and attached with additional adapter sequences for priming the next NGS steps (Appendix A). Multiplexing indices and sequencing adaptors were added by an additional PCR using the protocol supplied from the sequencing company, Macrogen (Seoul, South Korea). High-throughput sequencing was carried out with illumina Mini-seq (San Diego, CA, USA) with the paired-end multiplexed library. NGS raw reads from paired-end Mini-seq sequencing were joined by the program, ‘fastq-join’ implemented in the package, ‘eu-util’ (accessed on 21 October 2021, https://github.com/ExpressionAnalysis/ea-utils). The In/del efficiency was calculated using the program ‘Cas-Analyzer’ [44] (accessed on 21 October 2021, http://rgenome.net) or an in-house python script. All read counts and efficiencies from transgenic individuals can be found in Appendix A.

### 4.5. Chlorophyll Isolation and Measurement to Test Osmotic Tolerance to Salt Stress

To test osmotic tolerance to salt stress, T_4_ seeds were stratified at 4 °C for 3 days to maximize germinations synchronized and placed on the filter paper in the squared plastic container with Murahige and Skoog (MS) agar medium. Each MS plate was prepared with half-strength MS medium including vitamins (Duchefa Biochemie, Cat#M0222), 0.8% (*w*/*v*) agar (Duchefa Biochemie, Cat#P1001), and 2% sucrose (Duchefa Biochemie, Cat#S0809). All T_4_ seeds were arranged by grouping according to mutant lines (e.g., samples from the same genotype parent in two genes, *AT5G46510* and *ACQOS*), but the positions of seedlings belonging to grouped lines were randomly selected for the position in the same plate (Figure 2B). Each mutant line with allelic genotypes (wild-type or mutant alleles for *AT5G46510* and *ACQOS*) was prepared with 12 seeds. After 14 days after sowing, most of seeds were germinated and filter papers with these seedlings were transferred to new MS plate with the addition of 250 mM NaCl for short term acclimation for 14 days. All seedlings grown on the media with NaCl were again transferred into the MS plate added with 700 mM sorbitol. Seedlings grown on the MS plate with 700 mM sorbitol for 7 days were subject to extract chlorophylls. To extract chlorophylls from seedlings, we collected the aerial parts of seedlings and used the Dimethyl sulfoxide (DMSO) extraction method previously reported [45] with some modifications. First, we ground the seedlings after quick freezing with liquid nitrogen and added 1.0 mL of DMSO and mixed well to react for 2 min by placing the tube on the automated shaker. Mixed samples were centrifuged at 12,000 rpm for 5 min and we collected the supernatant. Next, we added 0.45 mL of DMSO to samples collected in the previous step and mixed for 2 min on the automated shaker. The supernatant of the reaction was collected after 5-min centrifugation at 12,000 rpm, and we added 0.45 mL of DMSO one more time and mixed for 2 min in the shaker, then collect the supernatant after 5-min centrifugation at 12,000 rpm. Finally, we obtained 1.9 mL of supernatant and used 1.0 mL to measure the absorption at the wavelength of A_645_ and A_663_ in the spectrophotometer (Optizen 1412V, Mecasys, Korea). To determine the chlorophyll amounts, we used Arnon’s equation [46] using measurement of absorption at A_645_ and A_663_ (Equations (1)–(3));
(1)$$\mathrm{Chlorophyll}\;a\;\left(g/l\right)=\frac{1000\cdot \left(0.0127\cdot a-0.00269\cdot b\right)}{W}$$
(2)$$\mathrm{Chlorophyll}\;b\;\left(g/l\right)=\frac{1000\cdot \left(0.0029\cdot a-0.00468\cdot b\right)}{W}$$
(3)$$\mathrm{Total}\;\mathrm{Chlorophyll}\;\left(g/l\right)=\frac{1000\cdot \left(0.0202\cdot a+0.00802\cdot b\right)}{W}$$

In the equation, ‘a’ refers the measure of A_663_, ‘b’ A_645_, and ‘W’ sample fresh weight (mg). Total chlorophyll amounts from each allelic mutant were compared with each other and with wild type Col-0 and a natural accession Bur-0 that belongs to haplogroup 4 (*AT5G46510* and *ACQOS* are deleted) and that showed a strong salt stress resistance in the previous studies [34,35]. Additionally, to reduce the possible bias from maternal effect or growth variation, we normalized the total chlorophyll amounts with those from untreated seedlings. All statistical tests of chlorophyll content comparisons were done with one-way ANOVA with *post hoc* Tukey HSD test in R program.

## Figures and Tables

**Figure 1 ijms-22-11389-f001:**
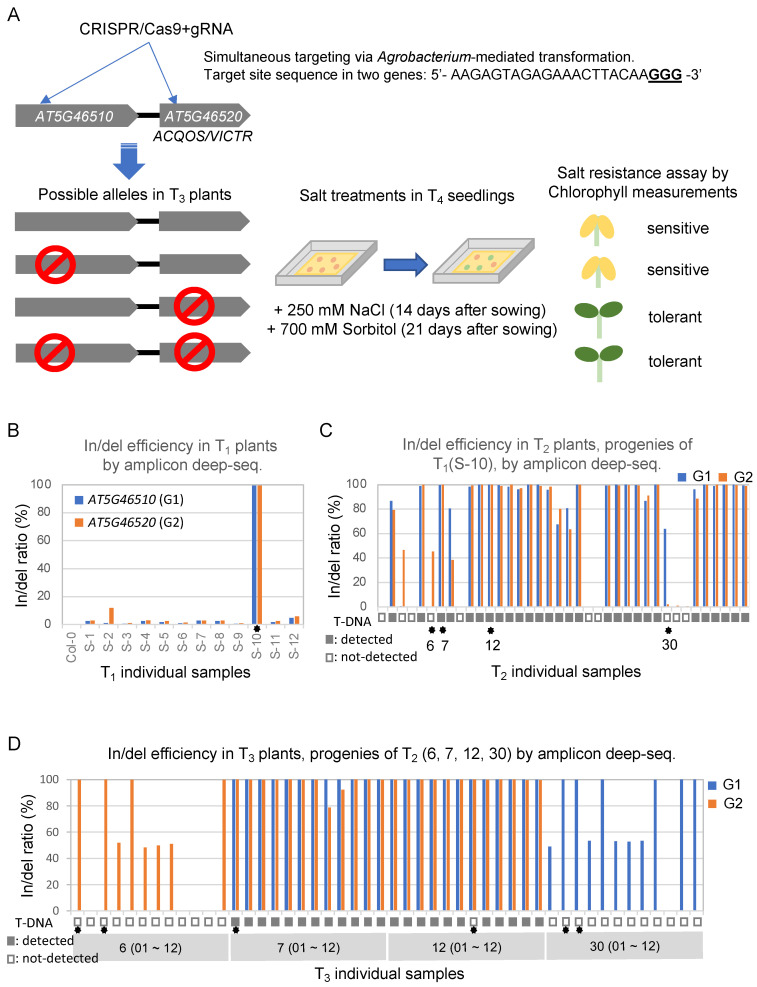
A schematic diagram depicting the study design and the insertion/deletion (in/del) efficiencies of individual transformants generated by CRISPR-mediated multiple-target mutagenesis in *Arabidopsis*. (**A**) A schematic illustration depicting the flow of this study. A single single-stranded guide RNA (sgRNA) that simultaneously targets two genes, *AT5G46510* (G1) and *AT5G46520* (*ACQOS*/*VICTR*; G2), as well as *Sp*Cas9, were expressed in *Arabidopsis* plants following *Agrobacterium*-mediated transformation to introduce in/del mutations. Transgenic seedlings in the fourth generation (T_4_) with desirable allele combinations confirmed in T_3_ parental plants were subjected to salt stress. Salt stress sorbitol treatments were applied to the seedlings of transgenic plants of the fourth generation (T_4_) for 7 days, followed by 7-day acclimation to sodium chloride (NaCl). Then, the chlorophyll contents of aerial tissues harvested from 28-day-old seedlings were measured as an indicator of salt tolerance. (**B**–**D**) In/del mutation efficiencies of individual transgenic plants of the first (**B**), second (**C**), and third (**D**) generations measured by amplicon deep sequencing for *AT5G46510* (G1) and *ACQOS* (G2). Transformants containing traces of T-DNA are indicated with closed (detected) or open (not-detected) squares. Asterisks indicate the mutant lines taken forward to the next generation in (**B**,**C**) and the parental lines of T_4_ seedlings in (**D**) used in downstream experiments.

**Figure 2 ijms-22-11389-f002:**
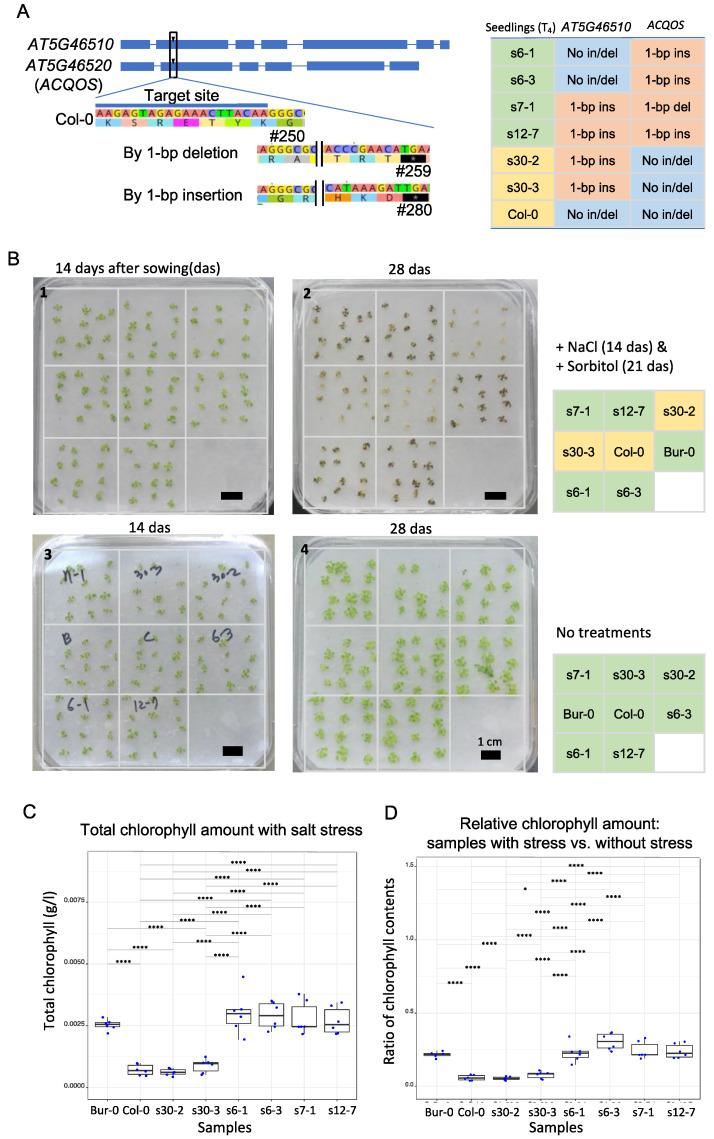
Salt stress resistance was determined in T_4_ transgenic plants by measuring chlorophyll content. (**A**) An illustration showing the mutagenesis target locations in *AT5G46510* and *AT5G46520: ACQOS*, as well as the predicted early stop codon positions for 1 bp deletion (top) or insertion (bottom) mutations. Table (right) showing the mutant alleles examined in the T_4_ transgenic lines. (**B**) Representative images of a batch of T_4_ seedlings. Seedlings (12 × 8 samples) of each genotype were arranged on filter paper within the culture plate and treated with sodium chloride (NaCl) at 14 days after sowing (das), then with sorbitol at 21 das (B1 and B2). Culture plates containing untreated control plants are shown for comparison (B3 and B4). Black bars are 1 cm scale bar. (**C**) Total chlorophyll contents of each genotype, as calculated by measuring the absorption at A_663_ and A_645_ and then applying Arnon’s equation. Each dot within the boxplots represents the chlorophyll content of ~12 pooled seedlings. One-way ANOVA with post hoc Tukey HSD test was performed to calculate significant differences between samples, as indicated with asterisks (**** *p* < 0.0001). (**D**) The relative chlorophyll contents of seedlings subjected to salt stress. Chlorophyll levels were normalized to those of untreated seedlings. Significant differences (*p*-values of 0.01–0.05) are indicated with an asterisk.

## Data Availability

All NGS data and in-house python script are available upon request to corresponding author.

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
