# Peer review of "The Functional Association of ACQOS/VICTR with Salt Stress Resistance in Arabidopsis thaliana Was Confirmed by CRISPR-Mediated Mutagenesis"

_ijms, 2021, doi:10.3390/ijms222111389_

Round 1

Reviewer 1 Report

In this manuscript, the authors did the functional association of ACQOS/VICTR with salt stress resistance in Arabidopsis thaliana that was confirmed by CRISPR-mediated mutagenesis. The author introduced small insertion/deletion mutations into two target genes using a single sgRNA and obtained transgenic plants with three different combinations of mutant alleles. The authors then tested our mutant lines for osmotic tolerance after a low-salt-stress acclimation period by determining the chlorophyll contents of the mutant seedlings. These results indicated that ACQOS is directly associated with salt resistance, while the neighboring NLR is not. Here, the authors confirmed previous findings suggesting the involvement of ACQOS in salt tolerance and demonstrated the usefulness of CRISPR-mediated mutagenesis in validating the functions of genes in a single genetic background.

The manuscript is very well written and has a new novel finding, however, I have my reservations since the study lack finding the mechanism for salt stress tolerance or some connection between the genotype and phenotype. The study also has only two figures. It would be better if the author checks salt stress tolerance in the mature plants not just in the salt medium. I would recommend author submit this study as a Brief report or short communication. 

I have found some plagiarized lines in this article at L29-30, L64-65, and from 258-288 (“The efficacy of CRISPR-mediated cytosine base editing with the RPS5a promoter in Arabidopsis thaliana”)

Author Response

I would like to thank you for your valuable comments and suggestions. 

Reviewer 2 Report

The reviewed article requires significant reworking in order to be at the standard suitable for publication consideration, namely (1) the wording of many sentences must be improved in order for each of these problematic sentences to make sense, and (2) a lot more background information is required throughout the manuscript to provide context to the reader. The lack of background information makes the entire article feel very rushed, and lacking completeness.

Along these lines, the entire article is very brief. Significant additional text is required to provide context to each experiment and/or explanation of each set of obtained results. Currently, the authors are relying on the reader having an extremely in-depth understanding of all experimentation and/or experimental approaches used and this approach greatly limits the impact, and therefore, the audience readership of the article. The provision of extensive additional background information / explanatory text would greatly improve the impact of the study which is a requirement of publication in a journal which carries an impact factor as high as that of IJMS.

I have attached an annotated PDF of my review of the article for the author’s reference which identifies all problematic sentences, incorrectly used words, and where additional background text is required for article improvement. The authors are requested to use this resource when preparing the revised version of their originally submitted manuscript.

Other issues are:

(1) why only two main text Figures provided?

(2) detailed phenotypic assessment of the newly generated plant lines is required, as is for all newly generated plant lines – why is this missing?

(3) why were so few plant lines used in the study?

(4) why so few biological replicates used in this study?

(5) the stress treatment regime used in this study is highly curious, therefore, why was it used instead of a standard approach for salt stress treatment of Arabidopsis?

(6) why was just a standard t-test used for the statistical assessment of the obtained results?

Author Response

(The authors gave the same response as above.)

Reviewer 3 Report

In this study, the authors applied (CRISPR)-mediated mutagenesis for simultaneous targeting of duplicated genes (At5G46520 and At5G46510) using a single sgRNA. This approach allowed the authors to generate three different mutant genotypes: carrying a mutation in one gene only and double mutant. 

In my opinion the current version of the manuscript cannot be published as a research article. The research is technically correct however I do find the current manuscript lacking in sufficient novelty of the methods and results. 

The approach to simultaneously modify homologous gene copies using single sgRNA for CRISPR/Cas9-mediated targeted mutagenesis has been successfully applied in many models (e.g in Rice by Endo et al. 2014 (Plant&Cell Physiology).

Only one physiological assay has been performed on mutants generated in this study, it was acquired osmotolerance assay previously described by Ariga et al. 2017 (Nature Plants). The results of the assay for CRISPR/Cas-9 mutant of ACQOS were similar to those obtained for SALK acqos mutants as previously described.

Moreover there is a number of issues that require substantial alteration to improve the quality of the manuscript or rather prepare as short communication. 

Line 20: should be “high salt-stress “ not “low salt-stress acclimation period” when applying 250 mM NaCl, moreover this information should be included in Figure or Figure captions.

Line 76: please add the appropriate reference

It is not clear what are “traces of T-DNA” , why there are monitored, which primers were used in PCR presented in Supplementary Figure S1?

Line (171-173) the authors should remove the description of the method from the Result section.

It is not explained why Bur-0 Arabidopsis accession was used in osmotolerance assay.

Line 187-206: The first part of the Discussion is very confusing. What is the new, different of innovative issue of this study? 

Line 232 – 235 this information belongs to the Materials and Methods section, the discussion should not have information that belongs to other sections.

Author Response

(The authors gave the same response as above.)

Round 2

Reviewer 1 Report

I am happy with the author's response. However manuscript is too short to be accepted as an original article. However, It can be accepted as short or brief communication.

Author Response

Response to Reviewer1.

I am happy with the author's response. However manuscript is too short to be accepted as an original article. However, It can be accepted as short or brief communication.

>RESPONSE: We would like to your constructive comments and suggestions. This manuscript will be categorized as a short ‘communication’.

Reviewer 2 Report

Dear authors,

Thank you for so thoroughly addressing my concerns with you original manuscript version.

The main issue that remains is the requirement further English language improvement.

Please again carefully go through your manuscript to check and improve all sections.

Author Response

Response to Reviewer2.

Dear authors,

Thank you for so thoroughly addressing my concerns with you original manuscript version.

The main issue that remains is the requirement further English language improvement.

Please again carefully go through your manuscript to check and improve all sections.

>>RESPONSE: We would like to thank you for your constructive comments and suggestions. This manuscript was initially checked with English editing service as below. Also, we will carefully check the manuscript for publication.

The English in this document has been checked by at least two professional editors, both native speakers of English. For a certificate, please see:

http://www.textcheck.com/certificate/LZIPA9

Reviewer 3 Report

The authors improved their manuscript however there are some issues that should be clarified.

I sustain my opinion that this manuscript cannot be accepted as original article because it contains only two Figures that still need to be improved. In my opinion this manuscript should be re-edited as a short communication.

As I had mentioned before 250 mM NaCl applied to Arabidopsis seedlings creates high salt stress but not low-salt stress as is stated in the Abstract section in line 21.

The concentration of stress factors should be included in the scheme in Figure 1.

The graphs presented in Figure 2 section C differ in size, different font size was used to mark the statistically significant difference, please unify the appearance of the graphs.

Author Response

Response to Reviewer3.

The authors improved their manuscript however there are some issues that should be clarified.

I sustain my opinion that this manuscript cannot be accepted as original article because it contains only two Figures that still need to be improved. In my opinion this manuscript should be re-edited as a short communication.

>RESPONSE: First of all, we would like to appreciate for your valuable comments and suggestion.

This manuscript is re-edited as a ‘communication’, one of article types in IJMS.

As I had mentioned before 250 mM NaCl applied to Arabidopsis seedlings creates high salt stress but not low-salt stress as is stated in the Abstract section in line 21.

> RESPONSE: I should have rechecked that part. I made a mistake to remain it unchanged. My apology and I changed as you suggested in the text.

The concentration of stress factors should be included in the scheme in Figure 1.

> RESPONSE: I revised as you suggested.

The graphs presented in Figure 2 section C differ in size, different font size was used to mark the statistically significant difference, please unify the appearance of the graphs.

>RESPONSE: I revised as you suggested.